# Morphological and Functional Evaluation of Kodkod (*Leopardus guigna*) Oocytes After In Vitro Maturation and Parthenogenetic Activation

**DOI:** 10.3390/ani15203031

**Published:** 2025-10-19

**Authors:** Deyna Toledo-Saldivia, Alonso Cáceres-Hernández, Daniela Doussang, Camila Zapata-Rojas, Sebastián Vergara, Ingrid Carvacho, Fidel Ovidio Castro, Lleretny Rodriguez-Alvarez, Daniel Veraguas-Dávila

**Affiliations:** 1Escuela de Medicina Veterinaria, Facultad de Ciencias de la Vida, Universidad Andres Bello, Viña del Mar 2531015, Chile; d.toledosaldivia@uandresbello.edu (D.T.-S.); daniela.doussang@unab.cl (D.D.); 2Departamento de Ciencia Animal, Facultad de Ciencias Veterinarias y Pecuarias, Universidad de Chile, Santiago 8820808, Chile; caceresalonso22@gmail.com; 3Sustainability Research Center (CIS), One Health Institute, Universidad Andres Bello, Santiago 8370146, Chile; 4Laboratorio de Biotecnología Animal, Departamento de Ciencia Animal, Facultad de Ciencias Veterinarias, Universidad de Concepción, Chillán 3812120, Chile; czapata@udec.cl (C.Z.-R.); fidcastro@udec.cl (F.O.C.); llrodriguez@udec.cl (L.R.-A.); 5Laboratorio de Reproducción y Canales Iónicos, Departamento de Medicina Traslacional, Facultad de Medicina, Universidad Católica del Maule, Talca 3480112, Chile; sebastian.vergara@alu.ucm.cl (S.V.); icarvacho@ucm.cl (I.C.)

**Keywords:** in vitro embryo production, felid embryos, in vitro culture, wild felids, domestic cat

## Abstract

**Simple Summary:**

The kodkod (*Leopardus guigna*) is a small wild felid native to Chile, whose population is currently in decline. Assisted reproductive technologies (ARTs) offer valuable tools for the genetic preservation of endangered species, including the kodkod. Conservation strategies for the kodkod have included cryobanking and the generation of cloned embryos through interspecific somatic cell nuclear transfer (iSCNT). However, the reproductive physiology and embryo development of this species are poorly understood. Here, we applied optimized protocols developed for domestic cat oocytes to evaluate the in vitro oocyte maturation (IVM) and in vitro culture (IVC) of embryos from kodkod. Immature cumulus–oocyte complexes (iCOCs) were collected from a female kodkod, and their morphology and developmental competence were evaluated. Immature kodkod oocytes were smaller than domestic cat oocytes. Kodkod oocytes were able to mature in vitro and develop to the blastocyst stage following chemical activation. Our work demonstrates that the protocols used to manage domestic cat oocytes can be successfully adapted for IVM and in vitro embryo production (IVP) in the kodkod. Our research provides a foundation for improving ARTs aimed at conserving this endangered felid and enhancing our understanding of its reproductive biology.

**Abstract:**

The kodkod (*Leopardus guigna*) is a vulnerable wild felid native to South America whose population is steadily declining. ARTs offer valuable tools for the preservation of its genetic diversity. Our study provides the first evaluation of the morphological and functional acquisition of competence in kodkod oocytes using protocols previously established for domestic cat oocytes. In total, 29 iCOCs were obtained from the ovaries of a single juvenile female kodkod that deceased in a wildlife rehabilitation center. Based on morphological criteria, 13 oocytes were selected for IVM and subsequently evaluated for developmental competence following parthenogenetic activation (PA) and in vitro culture (IVC). Kodkod oocytes appear to be smaller and have a thinner zona pellucida compared to those of domestic cat oocytes. These kodkod oocytes demonstrated the ability to mature in vitro, underwent cleavage, and developed in vitro to the blastocyst stage by day 9. Here, we show that protocols to manage domestic cat oocytes and embryos can support kodkod in vitro oocyte maturation, activation, and in vitro embryo development. However, given that the results were obtained from a single individual and the protocols were tested in a limited number of oocytes, further studies involving additional specimens are essential to validate these observations and refine ART applications for kodkod conservation.

## 1. Introduction

The kodkod or guigna is a vulnerable felid species with a decreased population and a reduced number of mature individuals (fewer than 100,000) [1,2]. The kodkod population has been principally affected by fragmentation of its habitat caused by human activities such as hunting and native forest destruction, which negatively affect its genetic diversity [3]. Among ARTs, artificial insemination (AI) and in vitro embryo production (IVP) are key strategies for maintaining the genetic diversity of endangered species [4,5,6]. Several studies have applied these approaches to support kodkod conservation. Our research group has contributed by establishing cryopreserved fibroblast banks and conducting cell cycle analyses to assess their potential for iSCNT [7]. We also characterized mesenchymal stem cells from abdominal fat for regenerative applications and produced kodkod blastocysts through iSCNT using domestic cat oocytes [8,9]. Despite these advances, further studies are needed to improve embryo production efficiency and strengthen the use of these biotechnologies in kodkod conservation.

The information related to kodkod reproductive physiology, including embryo development, is limited, and it is assumed that it is similar to that of domestic cats. However, to date, no studies have reported hormonal parameters and gamete evaluation in the kodkod, with this being crucial information for the development and improvement of ARTs in this species. Kodkod blastocysts have been generated by iSCNT, but their developmental capacity was low, and the morphological quality of the blastocysts was poor compared to domestic cat blastocysts [8]. This could have been caused by many factors, but principally by the species-specific differences between the kodkod and domestic cats. The morphology and competence of kodkod oocytes and their early embryo development have not been described, due to the difficulty associated with gamete collection in this species.

IVM of oocytes has been successfully applied in domestic cats and several wild felid species, representing a valuable approach to rescue immature oocytes from endangered individuals after medical ovariohysterectomy or postmortem collection [10,11]. IVM rates vary widely among species, ranging from 0% to 62.5% in both large felids (*Pantherinae*) and small felids (*Felinae*) [10,11]. Fertilization of these oocytes has been achieved by in vitro fertilization (IVF) using homologous or heterospecific sperm and by intracytoplasmic sperm injection (ICSI) with fresh or cryopreserved sperm, producing viable embryos that can reach the blastocyst stage [12,13]. These findings demonstrate that immature oocytes recovered postmortem from wild felids can mature and develop in vitro, offering a useful model to study oocyte competence and support conservation efforts.

Parthenogenetic activation (PA) is an experimental approach that induces oocyte activation through artificial stimuli, mimicking the physiological Ca^2+^ increase triggered by the spermatozoon during fertilization [14]. Although parthenogenetic embryos (parthenotes) lack full-term developmental potential, they provide a valuable model for assessing oocyte competence and embryo culture conditions, which are critical for improving ART efficiency [15,16,17,18]. This approach is particularly relevant for endangered species, such as wild felids, where gamete collection and embryo production are often limited. Evaluating activation responses in oocytes from endangered species contributes to the refinement of protocols required for in vitro embryo culture and advanced reproductive techniques such as ICSI and SCNT, which could be used for the reproduction and conservation of these species [19,20,21,22,23,24,25]. Several studies have evaluated the PA of domestic cat oocytes [25,26,27]. These protocols have been successfully used in the generation of embryos by iSCNT from different wild felid species [28,29]. The adaptation of IVM and PA protocols developed for the domestic cat provides an opportunity to assess oocyte competence and optimize ARTs in endangered felids such as the kodkod. Eventually, the establishment of reliable in vitro embryo production systems could support ex situ conservation programs and the creation of genetic resource banks, contributing to the long-term preservation of kodkod genetic diversity.

In this study, our research group collected immature COCs from kodkod ovaries donated by a wildlife rehabilitation center from a deceased individual. We hypothesized that the kodkod oocytes would be able to mature in vitro and generate embryos following parthenogenetic activation, using protocols previously adapted for the domestic cat. The objective was to evaluate the morphological characteristics and competence of kodkod oocytes subjected to in vitro maturation and parthenogenetic activation by using protocols previously adapted for domestic cat oocytes. The results of this study will contribute to the knowledge of oocyte morphology, competence, and embryo development in the kodkod, being helpful for the improvement of ARTs in this species.

## 2. Materials and Methods

All chemical reagents were purchased from Sigma Aldrich Chemicals Company (St. Louis, MO, USA), except for those otherwise indicated.

### 2.1. Ethics Statement

All procedures involving animal manipulation were approved by the ethics committees (Comité de Bioética de la Facultad de Ciencias Veterinarias de la Universidad de Concepción, certificate of approval CBE-082020; and the Comité Institucional de Cuidado y Uso de Animales, CICUA, of the Facultad de Ciencias Veterinarias y Pecuarias de la Universidad de Chile, 25902-VET-UCH).

### 2.2. Animals

Ovaries and testicles from healthy female and male domestic cats aged between 5 months and 1 year were collected for in vitro embryo production.

A pair of ovaries from a juvenile female kodkod were donated by a wildlife rehabilitation center (Centro de Rehabilitación de Fauna Silvestre de la Universidad de Concepción, CRFS-UdeC). The ovaries were obtained during the necropsy of this kodkod specimen that died from medical complications after being treated in the rehabilitation center due to a wild dog attack. The individual was delivered to the wildlife rehabilitation center in poor medical condition. When it died after medical intervention, the ovaries were donated after necropsy within the same day.

### 2.3. Experimental Design

The kodkod ovaries were received, and the iCOCs were collected from these by slicing. The kodkod iCOCs were morphologically classified using the parameters previously described for domestic cat iCOCs. IVM was performed using only grade I and II iCOCs from the kodkod, with domestic cat iCOCs used as a control during these procedures. The cumulus expansion was evaluated by measuring the diameter and area of each individual COC before and after IVM. Then, cumulus cells were removed and the maturation rate was estimated by visualization of the first polar body. The morphology of the kodkod and domestic cat oocytes was evaluated by measuring the following three parameters: the zona pellucida thickness (ZPT), oocyte cytoplasm diameter (OCD), and total oocyte diameter (TOD) (zona pellucida, perivitelline space, and oocyte). To evaluate oocyte competence, the matured oocytes from the kodkod were subjected to PA and cultured in vitro (IVC) using protocols designed for domestic cat oocytes. Additionally, domestic cat embryos were generated by PA, and IVF was used to validate these results. During IVC, the cleavage, morula, and blastocyst formation rates were evaluated in the kodkod and domestic cat groups. Finally, blastocyst staining was performed for total cell counting (Figure 1).

### 2.4. Ovariohysterectomy and Cumulus–Oocyte Complex Collection

Ovaries from female domestic cats were collected by ovariohysterectomy. The anesthesia protocol previously described by our research group was used in this study [8]. Ovaries from a kodkod were recovered after necropsy of a deceased individual. In both cases, ovaries were transported to the laboratory in a sterile 0.9% NaCl solution supplemented with 0.1% gentamycin, at approx. 38 °C.

iCOCs were recovered from ovaries by slicing into a 100 mm Petri dish containing 10 mL of medium 199 with Earle’s salts supplemented with 0.18 mM HEPES, 5% FBS, and 50 μg/mL gentamycin (He199) at 38.5 °C. The iCOCs collected were morphologically classified before IVM.

### 2.5. Morphological Classification of Immature Cumulus–Oocyte Complexes

Immature COCs from the kodkod and domestic cats were morphologically classified in grade I, II, III, and IV, with only the grade I and II COCs selected for in vitro maturation, as previously described by Wood et al. [30]. Grade I (excellent): Oocytes exhibit a uniform, dark cytoplasm and are surrounded by five or more layers of tightly compacted cumulus cells. Grade II (good): Oocytes display a uniform, dark cytoplasm with a complete corona radiata, but fewer than five layers of cumulus cells. Grade III (fair): Oocytes show a mosaic appearance of the cytoplasm, with nearly a full corona radiata and some cumulus cells that are not tightly compacted. Grade IV (poor): Oocytes present severe cytoplasmic mosaicism or fragmentation, accompanied by a sparse corona radiata and few cumulus cells; some oocytes are nearly denuded [30]. To evaluate the morphological quality of the kodkod and domestic cat COCs, each individual was considered a biological replicate, and the number and grading of COCs recovered from each felid were recorded.

### 2.6. In Vitro Maturation of Oocytes

In vitro maturation of iCOCs was performed as we previously described [8], using four-well dishes containing 500 μL of medium 199 with Earle’s salts, supplemented with 0.3% fraction-V BSA, 0.1 IU/mL FSH-LH (Pluset, Calier, Buenos Aires, Argentina), 1 μg/mL 17β-estradiol, 0.36 mM sodium pyruvate, 2 mM glutamine, 2.2 mM calcium lactate, 20 ng/mL EGF, and 50 μg/mL gentamycin (IVM-199) in a humidified gas atmosphere with 5.0% CO_2_ at 38.5 °C for 26 h. In total, 10 to 20 iCOCs were maturated per well.

### 2.7. Evaluation of Cumulus Cell Expansion

To evaluate the expansion of cumulus cells, pictures of the COCs were taken before and after IVM using a camera (Micrometrics CMOS Digital Camera, Accu-Scope, Commack, New York, NY, USA) attached to a stereomicroscope with high magnification (SZX16, Olympus Corporation, Santiago, Chile) and an image software (Micrometrics SE Premium software 4.5.1). Then, the diameter and area of each individual COC were measured using the software ImageJ version 1.54p (NIH, Rockville, MD, USA), as previously described by [31], with some modifications. The measurements were performed before (immature COCs) and after IVM (mature COCs) in the kodkod and domestic cat groups (Figure 2).

### 2.8. Morphological Evaluation of Oocytes

After IVM, the cumulus cells of COCs were removed using a 0.5 mg/mL hyaluronidase solution and vortex for 6 min. Then, pictures of the oocytes were taken using the method described before. The morphological evaluation of each oocyte was performed using the software ImageJ, as previously described by Kij-Mitka et al. [32], with some modifications, measuring the following three parameters: zona pellucida thickness (ZPT), oocyte cytoplasm diameter (OCD), and total oocyte diameter (zona pellucida, perivitelline space, and cytoplasm, TOD) (Figure 3).

### 2.9. Parthenogenetic Activation of Oocytes

The competence and in vitro embryo development of kodkod oocytes were evaluated by PA. This was performed due to the lack of kodkod sperm and to avoid the effects of heterospecific sperms. For PA, only mature oocytes with a visible polar body and without signs of degeneration were used. The chemical activation of kodkod oocytes was performed as previously described for domestic cat oocytes [33]. Matured oocytes were incubated in He199 with 7% ethanol, for 5 min, at 38.5 °C. Then, the oocytes were washed in medium-199 with ear salts, supplemented with 0.3% fraction-V BSA, 0.36 mM sodium pyruvate, 2 mM glutamine, 2.2 mM calcium lactate, and 50 μg/mL gentamycin, and incubated in this same medium with 10 µg/mL cycloheximide and 5 µg/mL of cytochalasin B for 5 h in a humidified gas atmosphere with 5.0% CO_2_, 5.0% O_2_, and 90% N_2_ at 38.5 °C.

### 2.10. Sperm Collection and In Vitro Fertilization

Because parthenotes are embryos generated by artificial methods, embryos produced by IVF were included in this study as biological controls of normal in vitro embryonic development in felids. Only male domestic cats older than 10 months of age were used as sperm donors. The anesthesia was performed as previously described [34]. Additionally, lidocaine (Lidocalm, 2%, Dragpharma, Santiago, Chile) was administered in the genital area as local anesthesia. Testes were transported in sterile 0.9% NaCl solution with 0.1% gentamycin at room temperature. The caudal portions of the epididymis were cut into small pieces of ~1 mm in a 100 mm Petri dish containing 10 mL of He199 supplemented with 0.3% fraction V BSA instead of FBS (He199-BSA) at 38.5 °C. Epididymal spermatozoa were processed and stored at 4 °C, as previously described [34]. IVF was conducted in four-well dishes with 500 μL of TALP medium supplemented with 6 mg/mL BSA, 0.36 mM sodium pyruvate, 1 mM glutamine, 2.2 mM calcium lactate, 1% MEM nonessential amino acids (NEAA), 0.5% MEM essential amino acids (EAA), 0.01 mg/mL heparin sodium salt, and 50 μg/mL gentamycin (TALP-IVF). Refrigerated spermatozoa were allowed to swim up for 30 min in He199-BSA at 38.5 °C [34]. The pellet was collected and resuspended in TALP-IVF. For IVF, 20 to 30 COCs were co-incubated with 1.5 to 2.5 × 10^6^ spermatozoa/mL in a humidified atmosphere of 5% CO_2_ in air at 38.5 °C for 24 h. Finally, cumulus cells were removed from the presumptive zygotes.

### 2.11. In Vitro Embryo Culture

To ensure development until the blastocyst stage, the activated oocytes derived from PA and the presumptive zygotes from IVF were in vitro cultured under specific conditions. The culture was performed in four-well dishes containing 500 μL of SOF medium supplemented with 0.37 mM trisodium citrate, 2.77 mM myo-inositol, essential and nonessential amino acids (final concentration 1×), 50 μg/mL gentamycin, 20 ng/mL EGF, and 3 mg/mL essentially fatty acid-free BSA (SOF-B) [34]. In total, 5–10 embryos were placed into each well. In vitro embryo culture was performed in a humidified gas atmosphere with 5.0% CO_2_, 5.0% O_2_, and 90% N_2_ at 38.5 °C, for 8 days. The cleavage was evaluated at day 2, morula formation at day 5, and blastocyst formation at day 8 of IVC.

### 2.12. Morphological Evaluation of Blastocysts

The morphological quality of the blastocysts was evaluated to confirm their formation and to compare morphological differences between blastocysts generated from PA of kodkod or domestic cat oocytes. In addition, IVF-derived blastocysts were used as a control representing normal in vitro development.

#### 2.12.1. Diameter Measurement

After in vitro culture, day 8 blastocysts were imaged using a camera mounted on a stereomicroscope, and the diameter of the blastocysts was measured from the captured images using the software ImageJ.

#### 2.12.2. Total Cell Counting

Blastocysts were fixed in a 3% glutaraldehyde solution for 72 h at 4 °C. Fixed blastocysts were stained with 5 μg/mL Hoechst 33342 (NucBlue™, ThermoFisher, Waltham, MA, USA) for 20 min and then were mounted on a slide and covered with coverslip. Visualization was achieved using the EVOS FL Auto Cell Imaging System (Thermo Fisher Scientific, Waltham, MA, USA) with a 20× magnification objective (200×).

### 2.13. Statistical Analysis

A *t*-test was used to evaluate significant differences in cumulus expansion and oocyte morphological measurements; this statistical test was chosen because metric data with a normal distribution were compared between two experimental groups (domestic cat and kodkod). The morphological evaluation of blastocysts among groups was performed using the same test, since the data are metric and normally distributed. Finally, the Kruskal–Wallis nonparametric test was used to evaluate in vitro embryo development among parthenogenetic activated oocytes from kodkod and domestic cats and in vitro fertilized oocytes; this was performed considering each individual oocyte as biological replicate. This nonparametric test was selected as the data are expressed as percentages with a non-normal distribution, and three experimental groups were analyzed. The statistical software InfoStat (2020 version; University of Cordoba, Córdoba, Argentina) was used to evaluate significant differences (*p* < 0.05).

## 3. Results

### 3.1. Morphological Classification of Cumulus–Oocytes Complexes Recovered from Kodkod Ovaries

The kodkod ovaries from one female were collected and processed by slicing to release iCOCs. A total of 29 iCOCs were obtained (Figure 4). From these, grade I COCs were not recovered. In total, 13 iCOCs were classified as grade II (44.8%) and selected from IVM. Six iCOCs were classified as grade III (20.7%) and ten as grade IV (34.5%); both groups (55.2%) were not considered for IVM (Table 1).

Additionally, domestic cat iCOCs were collected from 10 different female cats and were morphologically classified. In total, 489 were collected, with a mean of 48.9 ± 21.9 per cat. From these, 226 (46.2%) were considered suitable for IVM. Of the rest, 263 were classified as grade III and IV (53.8%) (Table 1).

### 3.2. Evaluation of COCs Expiation in Kodkod and Domestic Cat After IVM

The cumulus expansion of individual COCs was measured before and after IVM. A significant expansion of cumulus cells was observed in the COCs from kodkod and from domestic cats after IVM (*p* < 0.05) (Table 2).

### 3.3. Assessment of Kodkod Oocyte Maturation and Morphological Comparison Against Domestic Cat Oocytes

After IVM, the cumulus cells were removed and the visualization of the first polar body was performed to classify oocytes as mature (MII) or immature in the case that this was absent. In the kodkod group, 13 grade II COCs were subjected to IVM, and among these, 5 oocytes had a visible polar body (38.5%) and 8 oocytes did not mature after IVM (61.5%) (Figure 5). This demonstrates that kodkod oocytes are capable of maturing in vitro using a protocol originally designed for domestic cat oocytes. Although the maturation rate may appear low, it could be attributed to species-specific requirements for kodkod oocyte maturation.

No significant differences were observed in ZPT, OCD, and TOD between kodkod immature and MII oocytes, and between domestic cat immature and MII oocytes. No statistical differences were observed in OCD between kodkod and domestic cat MII oocytes. However, the ZPT tended to be significantly thinner in kodkod MII oocytes compared with domestic cat MII oocytes (*p* = 0.0628). The TOD of kodkod immature oocytes was significantly smaller than that of immature domestic cat oocytes (*p* = 0.0013). Similarly, the TOD of kodkod MII oocytes was smaller than that of their domestic cat counterparts (*p* = 0.0003). When the results of immature and mature oocytes were combined, the ZPT of kodkod oocytes was significantly thinner than that of domestic cat oocytes (*p* = 0.0249). Furthermore, the TOD of kodkod oocytes was significantly smaller than that of domestic cat oocytes (*p* < 0.0001) (Table 3).

### 3.4. Evaluation of In Vitro Development of Kodkod and Domestic Cat Oocytes After Parthenogenetic Activation

This experiment was conducted to evaluate the competence of in vitro matured kodkod oocytes, previously collected from one individual. Five kodkod oocytes with a visible polar body were activated, and after 48 h, 100% cleavage was observed. At day 5 of IVC, two morulae were observed, two embryos were at eight-cell stage, and one was at four-cell stage. Finally, no blastocysts were visualized on day 8. However, a single blastocyst was observed at day -9, probably formed at day 8.5 of IVC (Figure 6).

Additionally, domestic cat oocytes were activated using the same protocol. For statistical analyses, each activated oocyte was considered as an individual. A high number of morulae were observed at day 5 of IVC compared to the kodkod group, but these results were not significant (*p* = 0.19). Furthermore, a similar blastocyst rate was obtained between both groups (*p* = 0.91). Domestic cat embryos generated by IVF were included in this experiment to compare the in vitro developmental capacity against parthenogenetic embryos. Domestic cat IVF embryos showed a lower cleavage rate on day 2 and a higher blastocyst rate at day 8 than domestic cat PA embryos (*p* < 0.05) (Table 4). These results demonstrate that in vitro-matured kodkod oocytes are capable of developing in vitro after parthenogenetic activation up to the blastocyst stage. Despite the low number of activated oocytes, this developmental rate was similar to that observed in domestic cat oocytes.

### 3.5. Morphological Evaluation of Kodkod and Domestic Cat Blastocysts Obtained After Parthenogenetic Activation

The morphology of the single presumptive blastocyst obtained after PA of kodkod oocytes was evaluated to verify true blastocyst-stage formation, by comparing it with the morphology of domestic cat blastocysts generated after PA and IVF.

Despite the fact that only one blastocyst was obtained in the kodkod group and no proper statistical analyses could be performed, its diameter and total cell number were similar to the domestic cat blastocysts generated by parthenogenetic activation (Figure 6). Additionally, in this experiment, domestic cat blastocysts generated by PA had fewer cells than blastocysts generated by IVF (*p* < 0.05), but no differences were observed in blastocyst diameter (Table 5). However, in the only kodkod blastocyst produced, a defined inner cell mass (ICM) was not clearly appreciated, and it seems that a higher nucleus fragmentation was observed after Hoechst staining compared to domestic cat blastocysts generated by PA. Several domestic cat blastocysts generated by PA did not have a defined ICM, and this was more clearly appreciated in IVF-produced blastocysts (Figure 7). This indicates that a blastocyst was generated after kodkod in vitro matured oocyte activation, but this had a poor morphological quality compared to domestic cat blastocysts, especially to those generated by IVF (Figure 7).

## 4. Discussion

This study provides the first evidence that kodkod oocytes can mature and be parthenogenetically activated using protocols established for domestic cats. These results suggest that molecular mechanisms regulating oocyte competence and early development are conserved between these felids. The delayed blastocyst formation in kodkod parthenotes (day 9 vs. day 6 in cats) may reflect species-specific embryo kinetics, metabolic differences, or suboptimal culture conditions not yet adapted to kodkod requirements. The smaller size and thinner zona pellucida observed in kodkod oocytes may also influence metabolic exchange and developmental potential. Together, these features indicate that domestic-cat-based protocols are a useful foundation but require optimization to improve developmental competence. Refining in vitro culture systems and activation methods according to species-specific physiology will enhance ART efficiency in kodkods. Despite current limitations due to sample availability, this pioneering study establishes a framework for future reproductive research and conservation of this endangered felid.

The kodkod has been described as a seasonal long-day polyestrous species, similar to the domestic cat [35]. Its longer gestational period (72–78 days vs. 56–69 days in domestic cats) [36,37,38] suggests distinctive endocrine and metabolic adaptations. Genetic divergence within the *Leopardus* genus, characterized by a 36-chromosome karyotype (34 autosomes plus XX or XY), contrasts with the 38-chromosome configuration typical of most felids, including the domestic cat [39,40]. Such chromosomal differences may influence meiotic behavior and genome activation, requiring adjustments to IVM or IVC protocols. However, hybridization studies indicate that this divergence does not prevent compatibility among *Leopardus* species and domestic cats. *Leopardus geoffroyi* can produce viable hybrids with 37 chromosomes [41,42], and *L. geoffroyi* sperm can fertilize domestic cat oocytes in vitro, generating blastocysts [43]. These findings suggest that, despite genetic divergence, key reproductive mechanisms are conserved, supporting the adaptation of domestic-cat-based ARTs for kodkod reproduction.

Information regarding IVM of oocytes from wild felids remains scarce. In most cases, oocytes are collected in vivo after gonadotrophin stimulation and laparoscopic aspiration rather than matured in vitro [4,44,45], since IVM-derived oocytes often display altered transcriptomic profiles and reduced developmental competence compared to in vivo matured oocytes [46,47]. The reported IVM rates in felids (20–60%) are generally lower than those of ruminants (80–90%) [48,49,50,51]. Nevertheless, IVM remains the only feasible approach to recover iCOCs when endangered felids die unexpectedly or undergo ovariohysterectomy [11,12]. In this study, only 44.8% of kodkod iCOCs were suitable for IVM, consistent with the low morphological quality usually reported in prepubertal wild felids [52]. Despite this limitation, significant cumulus expansion and a 38.5% maturation rate were achieved, supporting the efficiency of the applied protocol. The measurement of cumulus expansion has been widely described in bovine COCs, where increased expansion has been associated with increased oocyte maturation and blastocyst formation [32,53,54]. These results reveal that the kodkod oocyte retains a functional competence comparable to other wild felids, suggesting conserved molecular mechanisms regulating meiotic resumption. These findings demonstrate that domestic-cat-based protocols can serve as a foundation for developing species-specific ARTs in *Leopardus* species. This increases our knowledge about oocyte biology among felids and provides a practical framework for adapting IVM protocols to improve reproductive efficiency and conservation outcomes in the kodkod.

The in vitro development of kodkod embryos has previously been assessed only through iSCNT using enucleated domestic cat oocytes [8]. These embryos showed a low developmental capacity compared with domestic cat clones, likely due to the limited ability of domestic cat oocytes to reprogram kodkod somatic nuclei. Kodkod cloned embryos were arrested at the morula stage, and only 5.6% reached the blastocyst stage after aggregation [8]. In this study, despite the limited number of activated oocytes, a 20.0% blastocyst rate was obtained, suggesting possible species-specific differences compared to the previous results obtained after iSCNT. During iSCNT, reprogramming factors within the oocyte cytoplasm reset the somatic nucleus [55,56]. However, cloning efficiency decreases as the phylogenetic distance between the donor nucleus and the oocyte donor species increases [29,57].

Oocyte competence is closely associated with morphological quality, as structural differences influence the cytoplasmic environment and reprogramming potential [58,59]. In this study, kodkod oocytes were significantly smaller and had a thinner zona pellucida than domestic cat oocytes, revealing species-specific traits not previously reported in this species. In domestic cats, oocyte lipid content and morphology correlate with energy metabolism and maturation efficiency [60,61], while larger oocytes with thicker zonae pellucidae show greater developmental and hatching potential [33]. The zona pellucida also mediates molecular communication; its removal in domestic cat blastocysts alters pluripotency and trophectoderm gene expression, miRNA release, and proteomic profiles [35,62,63,64]. Similar associations between zona characteristics and developmental success occur in rabbits, mice, and humans [65,66,67,68]. These findings suggest that the thinner zona pellucida and smaller oocyte size in kodkods may reflect species-specific differences in early embryonic development compared to domestic cat oocytes. These differences should be taken into consideration to improve IVM and IVC protocols for embryo production and the genetic preservation of the kodkod.

PA has been widely used to assess oocyte competence and culture conditions in wild species [24]. In domestic cats, PA-derived embryos showed a reduced developmental potential compared with IVF embryos [69,70], but higher blastocyst rates than those obtained by SCNT or iSCNT due to reprogramming failures of the cloning process [68]. In this study, a kodkod blastocyst was produced after PA at a rate comparable to domestic cat oocyte activation, representing the first evidence of in vitro embryo development in the kodkod. Although in this study a limited number of oocytes were obtained from a single individual, these findings expand current understanding of oocyte competence and developmental kinetics in the kodkod. The delayed blastocyst formation (day 8.5–9) observed in this species is similar to that reported for lion ICSI embryos [13], and it may reflect species-specific embryonic timing rather than suboptimal conditions. This study provides a basis for future comparative analyses and protocol optimization in endangered *Leopardus* species. Understanding reproductive biology is a critical component of species conservation. It is also a prerequisite for the development of ARTs, which have great potential to overcome infertility issues and maintain genetic diversity in rare and endangered species [71,72].

## 5. Conclusions

In conclusion, this study addresses a major knowledge deficit in the reproductive biology of the kodkod, providing the first evidence that its oocytes can mature, be chemically activated, and develop in vitro until the blastocyst stage, using protocols adapted for domestic cat oocytes. These findings reveal conserved mechanisms of oocyte competence among these felids. Despite these results being limited to postmortem oocytes recovered from a single individual, this work establishes a foundation for future studies. Expanding sample size and optimizing species-specific culture systems will enhance IVP efficiency and support kodkod conservation through advanced reproductive approaches.

## Figures and Tables

**Figure 1 animals-15-03031-f001:**
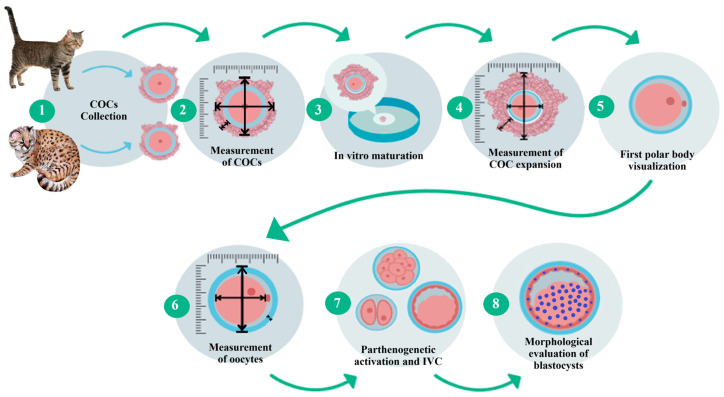
Experimental design. Step-by-step sequence of the methodology carried out for domestic cat and kodkod oocyte morphological and functional evaluation. Representative images.

**Figure 2 animals-15-03031-f002:**
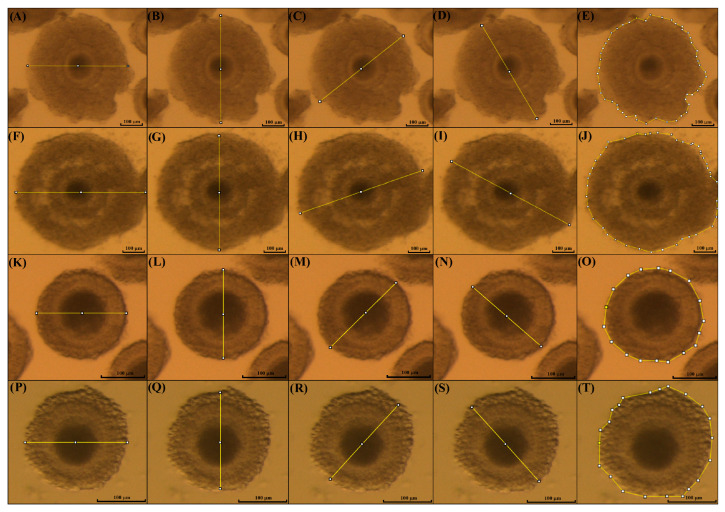
Measurement of COCs’ expansion after IVM. Measurement of COCs using the software Image J 1.54p. The mean of four different measures was used to estimate the perimeter of domestic cat iCOCs before IVM (**A**–**D**), mature COCs after IVM (**F**–**I**), and in kodkod COCs before (**K**–**N**) and after IVM (**P**–**S**). Furthermore, the area of immature and mature COCs was measured before (**E**) and after IVM, respectively (**J**) in domestic cats, and in the kodkod group (**O**) and (**T**), respectively.

**Figure 3 animals-15-03031-f003:**
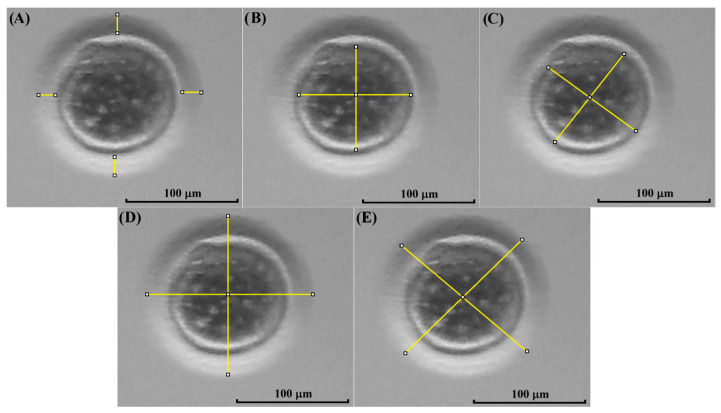
Measurement of kodkod and domestic cat oocytes. In this figure, a kodkod in vitro matured oocyte is shown, indicating the different measurement parameters used. The mean of four different measures was used to estimate the zona pellucida thickness (ZPT) using the software ImageJ (**A**). Similarly, four different measures were used to estimate the oocyte cytoplasm diameter (OCD) (**B**,**C**) and the total oocyte diameter (TOD) (**D**,**E**).

**Figure 4 animals-15-03031-f004:**
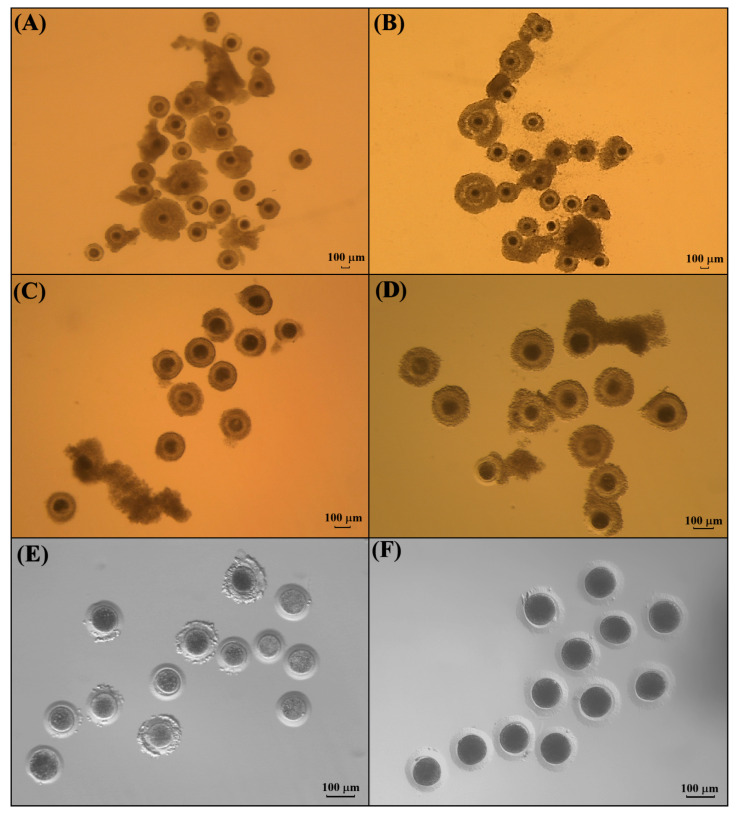
Morphology of COCs before and after IVM. (**A**) Grade I and II iCOCs from domestic cats. (**B**) COCs from domestic cats after IVM. (**C**) Grade II iCOCs from kodkod, selected for IVM. (**D**) COCs from kodkod after IVM. (**E**) Oocytes from kodkod after IVM and cumulus cells’ removal. (**F**) Oocytes from domestic cat after IVM and cumulus cells’ removal.

**Figure 5 animals-15-03031-f005:**
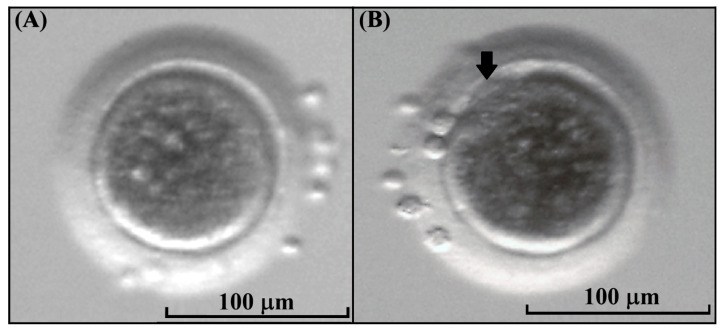
Kodkod oocyte morphology. (**A**) Kodkod oocyte without a visible first polar body. (**B**) Kodkod oocyte at the metaphase II stage with a visible polar body (arrow).

**Figure 6 animals-15-03031-f006:**
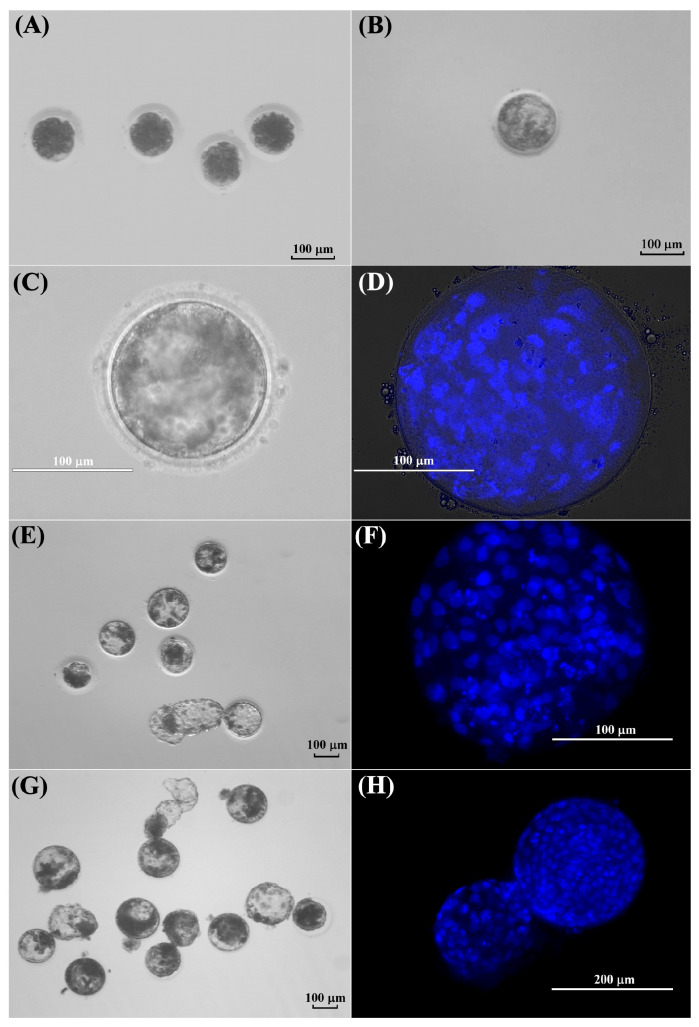
In vitro development of kodkod and domestic cat oocytes after parthenogenetic activation and in vitro fertilization. (**A**) Kodkod embryos generated by PA at day 5 of IVC. (**B**) Kodkod blastocyst obtained by PA at day 9 of IVC. (**C**) Kodkod blastocyst visualized using a transmitted light inverted microscope at day 9 of IVC. (**D**) Kodkod blastocyst covered with a coverslip and stained with Hoechst. (**E**) Domestic cat blastocysts generated by PA at day 8 of IVC. (**F**) Domestic cat blastocyst generated by PA stained with Hoechst. (**G**) Domestic cat blastocysts generated by IVF at day 8 of IVC. (**H**) Domestic cat blastocyst generated by IVF stained with Hoechst.

**Figure 7 animals-15-03031-f007:**
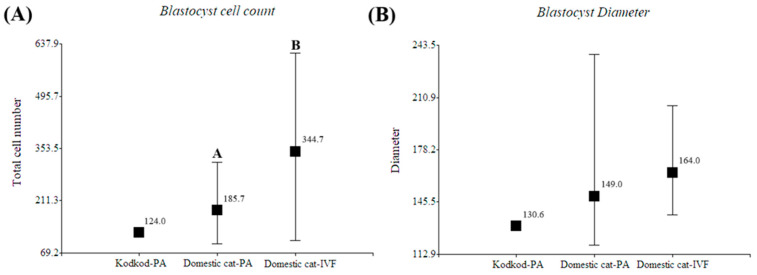
Morphological evaluation of in vitro produced blastocysts. (**A**) Blastocyst cell count (Mean; Min/Max) and (**B**) Blastocyst diameter measurement (Mean; Min/Max) of kodkod blastocyst (Kodkod—PA) and domestic cat blastocyst generated by parthenogenetic activation (PA) and in vitro fertilization (IVF). ^A,B^: indicates significant differences between groups in the same graphic.

**Table 1 animals-15-03031-t001:** Morphological classification of immature COCs collected from kodkod and domestic cat ovaries.

Group	N *	Total COCs	Grade I (%)	Grade II (%)	Grade I and II (%)	Grade III and IV (%)
Kodkod	1	29	0	13 (44.8)	13 (44.8)	16 (55.2)
Domestic cat	10	489	99 (20.3 ± 8.6)	127 (25.9 ± 5.4)	226 (46.2 ± 11.0)	263 (53.8 ± 10.9)

* Number of replicates in each experimental group; each replicate corresponds to one individual (female cat or kodkod).

**Table 2 animals-15-03031-t002:** Cumulus cell expansion (Mean ± SD) of kodkod and domestic cat immature (iCOCs) and mature cumulus–oocyte complexes (mCOCs), before and after IVM, respectively.

Group	N *	Diameter Measurement (µm)	Area Measurement (µm)
Kodkod iCOCs	13	170.8 ± 33.9 ^a^	21,042.9 ± 8198.2 ^a^
Kodkod mCOCs	13	222.7 ± 16.8 ^b^	35,874.5 ± 3921.2 ^b^
Domestic cat iCOCs	11	225.7 ± 40.3 ^b^	42,380.3 ± 16,451.3 ^c^
Domestic cat mCOCs	11	297.8 ± 85.4 ^c^	76,161.7 ± 42,536.0 ^d^

^a–d^: different superscripts indicate significant differences among groups. * N: indicate the number of COCs measured in each group.

**Table 3 animals-15-03031-t003:** Morphological measurement (µm) of kodkod and domestic cat oocytes after IVM (Mean ± SD).

Group	* Total Oocytes	Zona Pellucida Thickness (ZPT)	Oocyte Cytoplasm Diameter (OCD)	Total Oocyte Diameter (TOD)
Kodkod—Immature	8	19.5 ± 2.0	106.3 ± 6.6	147.0 ± 9.1 ^a^
Kodkod—MII	5	19.9 ± 0.9	108.8 ± 6.7	152.6 ± 3.7 ^a^
Total Kodkod	13	19.8 ± 1.2 ^A^	107.3 ± 6.5	149.1 ± 7.8 ^A^
Domestic cat—Immature	11	20.9 ± 2.9	111.5 ± 6.3	165.5 ± 11.2 ^b^
Domestic cat—MII	16	21.9 ± 2.3	109.4 ± 3.7	164.9 ± 5.8 ^b^
Total domestic cat	27	21.5 ± 2.5 ^B^	110.3 ± 4.9	165.2 ± 8.2 ^B^

^a,b^: different superscripts indicate significant statistical differences among immature and MII oocytes (*p* < 0.05). ^A,B^: different superscripts indicate statistical differences between total oocyte values (*p* < 0.05). *: Indicate the number of oocytes measured in each group.

**Table 4 animals-15-03031-t004:** Evaluation of in vitro embryo development (Mean ± SD) after parthenogenetic activation (PA) of mature oocytes from kodkod and domestic cat and in vitro fertilization.

Group	* N	Total Activated Oocytes	Cleavage N (%)	Morulae N (%)	Blastocysts N (%)
Kodkod—PA	1	5	5/5 (100)	2/5 (40.0)	1/5 (20.0) ^a^
Domestic cat—PA	5	56	53/56 (94.6 ± 6.9) ^a^	33/53 (62.3 ± 13.9)	11/53 (20.8 ± 4.5) ^a^
Domestic cat—IVF	5	138	56/138 (40.6 ± 20.8) ^b^	33/56 (58.9 ± 9.9)	18/56 (32.1 ± 7.7) ^b^

^a,b^: Different superscripts indicate significant differences in the same column. * Indicate the number of replicates performed in each experimental group.

**Table 5 animals-15-03031-t005:** Morphological evaluation of kodkod and domestic cat blastocysts obtained by parthenogenetic activation (PA) and in vitro fertilization (IVF).

Group	* N	Total Cell Number (Mean ± SD)	Total Diameter (µm) (Mean ± SD)
Kodkod—PA	1	124.0	130.6
Domestic cat—PA	10	185.7 ± 78.9 ^a^	148.9 ± 34.6
Domestic cat—IVF	10	344.7 ± 199.9 ^b^	164.0 ± 21.4

^a,b^: Different superscripts indicate significant differences in the same column. *: indicate the number of blastocysts evaluated in each experimental group.

## Data Availability

The data presented in this study are available upon request from the corresponding author.

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
