# Peer review of "Morphological and Functional Evaluation of Kodkod (Leopardus guigna) Oocytes After In Vitro Maturation and Parthenogenetic Activation"

_animals, 2025, doi:10.3390/ani15203031_

Round 1
Reviewer 1 Report
Comments and Suggestions for Authors
I have reviewed the article entitled “Morphological and functional evaluation of kodkod (Leopardus guigna) oocytes after in vitro maturation and parthenogenetic activation.” In this study, the authors evaluated the morphological and developmental characteristics of kodkod oocytes following in vitro maturation and parthenogenetic activation. The results indicated that kodkod oocytes were able to mature and develop to the blastocyst stage after PA using protocols established for domestic cat oocytes.
This study has some merit in contributing to the understanding of oocyte competence and embryo development in the kodkod. However, only 13 immature COCs were matured, and just 5 MII-stage oocytes were obtained in the present study. The authors should carefully consider whether this limited sample size is sufficient to support the conclusions.
Specific Comments:
- The Simple Summary is missing important results and should be revised to better reflect the main findings.
- In Figure 1, are the pictures of the cat and kodkod derived from the experimental materials, or are they representative images? Please clarify.
- Please define Grade I and Grade II COCs. Although the Methods section cites Wood et al. as a reference, the reviewer believes it is necessary to provide clear definitions within the manuscript.
- How many times was the experiment repeated? This information is essential for assessing reproducibility.
- Please carefully re-check the sentence in lines 266–267, as it may contain errors.
- In Table 1, what does “Grade I y II” mean? Please correct or clarify.
- In Figure 3, only domestic cat immature and mature COCs are shown, while kodkod COCs are missing. The reviewer believes that including images of kodkod COCs is more important than those of domestic cat COCs.
- As shown in Table 2, there were no significant differences in diameter and area measurements between domestic cat iCOCs and mCOCs. Does this suggest that the current maturation conditions may not be suitable for domestic cats? Please discuss.
- In Figure 4, please mark the position of the polar body with an arrow. Additionally, it appears that there are two polar bodies in the MII-stage oocyte—please clarify.
- In Figure 5, do the oocytes shown belong to domestic cats or kodkods? Please specify.
- In Table 4, is the data statistically significant? Since the Kodkod-PA group was based on only one experimental replicate, the statistical validity should be clarified.
- Some sections are unnecessarily wordy; a more concise style is recommended.
Author Response
I would like to thank the reviewers for their observations and comments, which helped improve the manuscript from different perspectives. Most of the suggestions were incorporated into this revised version of the manuscript, and those that were not have been duly justified. Below, you will find the responses to each of your comments.
Comments 1: This study has some merit in contributing to the understanding of oocyte competence and embryo development in the kodkod. However, only 13 immature COCs were matured, and just 5 MII-stage oocytes were obtained in the present study. The authors should carefully consider whether this limited sample size is sufficient to support the conclusions.
Response 1: Thank you for the observation. We are aware that the analysis includes oocytes from only one specimen. However, the kodkod is a highly vulnerable species, not found in zoos or conservation centers, and with a very limited population size. Therefore, the collection of oocytes and embryos from this species is extremely difficult and it has a high value. Therefore, we took the opportunity to share our findings with the scientific community, as this exceptional event provides valuable insights into the reproductive physiology of kodkod. We decided to try to publish our results despite the limited number of samples.
Considering this, the analysis presented in this manuscript focuses on kodkod oocytes rather than on the individual itself. The number of oocytes obtained allowed us to perform statistical analyses of their morphology and cumulus expansion, and also to observe the early stages of pre-implantation development in this species following oocyte activation. Our conclusions are general, and we acknowledge that additional samples are required to establish more robust findings.
We have clearly stated throughout the manuscript, including in the Simple Summary, Abstract, Introduction, Methods, Results, Discussion, and Conclusion sections, that further studies including a larger number of individuals and samples are required to better validate these findings.
Specific Comments and Responses:
Comments 1: The Simple Summary is missing important results and should be revised to better reflect the main findings.
- Response 1: I agree. The most important results of this study were incorporated into the Simple Summary. However, Reviewer 2 recommended explaining them in a simpler way for a non-specialized audience and placing greater emphasis on conservation. A rewrite of the Simple Summary was carried out in an attempt to incorporate both opinions.
Comments 2: In Figure 1, are the pictures of the cat and kodkod derived from the experimental materials, or are they representative images? Please clarify.
- Response 2: Yes, the pictures of the cat and kodkod are representative images. This clarification was also added into the figure description.
Comments 3: Please define Grade I and Grade II COCs. Although the Methods section cites Wood et al. as a reference, the reviewer believes it is necessary to provide clear definitions within the manuscript.
- Response 3: I agree. The definition of Grade I and II COCs was included in this new version.
Comments 4: How many times was the experiment repeated? This information is essential for assessing reproducibility.
- Response 4: The kodkod in vitro maturation and parthenogenetic activation was repeated once, because only one ovary pair were collected. In the domestic cat experiment. IVM was repeated 10 times, and IVF and PA were repeated 5 times each. This information was clarified in the text and table description.
Comments 5: Please carefully re-check the sentence in lines 266–267, as it may contain errors.
- Response 5: I agree. The error (xx) was corrected and the correct values were added.
Comments 6: In Table 1, what does “Grade I y II” mean? Please correct or clarify.
- Response 6: I agree. This was corrected “Grade I y II” was replaced by “Grade I and II”
Comments 7: In Figure 3, only domestic cat immature and mature COCs are shown, while kodkod COCs are missing. The reviewer believes that including images of kodkod COCs is more important than those of domestic cat COCs.
- Response 7: I agree. The images of kodkod COCs were included in this figure.
Comments 8: As shown in Table 2, there were no significant differences in diameter and area measurements between domestic cat iCOCs and mCOCs. Does this suggest that the current maturation conditions may not be suitable for domestic cats? Please discuss.
- Response 8: Additional data was added to validate cumulus expansion. In felid COC cumulus expansion is generally low, Cumulus cell expansion in feline COCs is generally low after IVM. This is mainly because many iCOCs have few cumulus cells at the time they are collected. Despite this, around 40–60% of oocytes usually reach the MII stage. This discussion was also added to the manuscript.
Comments 9: In Figure 4, please mark the position of the polar body with an arrow. Additionally, it appears that there are two polar bodies in the MII-stage oocyte—please clarify.
- Response 9: Only one PB is observed in the picture, it is a large one, and this size was observed also in other kodkod oocytes. The position was marked with an arrow. The other spherical structures are remaining cumulus cells attached outside the zona pellucida. I understand they could be mistaken for two polar bodies. Thank you for the observation.
Comments 10: In Figure 5, do the oocytes shown belong to domestic cats or kodkods? Please specify.
Response 10: In Figure 5, a kodkod oocyte was used as an example to illustrate the measurement parameters. This has been indicated in the revised version of the manuscript
Comments 11: In Table 4, is the data statistically significant? Since the Kodkod-PA group was based on only one experimental replicate, the statistical validity should be clarified.
- Response 11: For the statistical analysis, each embryo was considered as an individual, since only one replicate was performed in the kodkod group, in this explanation was included in the text.
Comments 12: Some sections are unnecessarily wordy; a more concise style is recommended.
- Response 12: The text was shortened across all sections, in accordance with this comment and with the suggestions made by Reviewer 2
Reviewer 2 Report
Comments and Suggestions for Authors
The study investigates the morphological and functional characteristics of Leopardus guigna (kodkod) oocytes, collected postmortem, with the aim of adapting domestic cat protocols for maturation and early embryonic development in this endangered species. The manuscript presents novel data regarding kodkod oocyte development to the blastocyst stage in vitro, a significant contribution given the lack of prior information on this species' reproductive biology. The comparative analysis with domestic cats is a logical approach, and the research addresses a critical gap in conservation efforts for the kodkod.
However, the manuscript, in its current form, requires substantial revisions to both its text and experimental methods before it can be considered for publication. While the topic is highly relevant and the initial findings promising, clarity, methodological rigor, and the interpretation of results need significant improvement. Therefore, my recommendation is Reconsider after major revision (substantial revisions to text or experimental methods needed).
Below are my specific comments and suggestions for revision:
- The manuscript presents groundbreaking information on kodkod oocyte biology and development, which is highly valuable for conservation. However, the presentation often obscures the novelty and significance of these findings.
- A critical limitation throughout the study is the use of a single kodkod individual and a small number of oocytes. This aspect needs to be explicitly and consistently highlighted and discussed in all relevant sections (Abstract, M&M, Results, Discussion, Conclusion).
- The overall narrative and logical flow need to be significantly improved across all sections to make the manuscript more coherent and accessible, particularly for an interdisciplinary audience that may include conservationists.
- Simple Summary:
- The current summary contains too much specialized terminology without adequate explanation for a non-specialist audience.
- It reads more like a condensed abstract rather than a simple, accessible summary.
- Please explicitly highlight the connection between this research and the protection/conservation of the kodkod species.
- Move detailed methodological descriptions to the abstract and/or methods section.
- Emphasize the unique opportunity and importance of working with oocytes from an endangered species.
- Provide brief, clear explanations for terms such as "blastocyst" and "polar bodies."
- The conclusive sentence needs to be rewritten to better convey the significance of this work as a foundational step for future kodkod conservation efforts.
- Abstract:
- The abstract currently blends methods, results, and conclusions, leading to a fragmented and difficult-to-follow narrative.
- Revise the logical flow to clearly separate background, objectives, methods overview, key results, and main conclusions.
- Avoid overly long sentences packed with multiple ideas. Strive for conciseness and clarity.
- Remove exact numerical data (mean ± SD) from the abstract; these details belong in the results section. Focus on the main findings and their significance.
- Crucially, explicitly state and emphasize that only one individual kodkod and a very limited number of oocytes (e.g., 13 suitable for maturation) were used. This is a major limitation that must be transparently acknowledged upfront.
- The current conclusion is too general. Clearly articulate the specific contribution of this research and explain why these initial results are important beyond simply being the first report.
- Introduction:
- The initial broad review of wild felids is too extensive and dilutes the focus on the kodkod. Begin the introduction by immediately highlighting the endangered status, population decline, and threats faced by the kodkod.
- Address the repetition of "assisted reproductive technologies" (ART). Organize the information more effectively, perhaps introducing ART generally, then focusing on specific techniques relevant to this study.
- Some references regarding reproductive technologies in felids appear to reiterate similar ideas. Streamline these discussions to avoid redundancy.
- The introduction currently reads too much like a general review of methods. Focus on building the specific context for this study.
- Clearly articulate the existing knowledge gap regarding kodkod reproduction early in the introduction. State explicitly that "little is known on kodkod reproduction."
- Strengthen the link between reproductive technologies and their role in kodkod conservation strategies.
- The research goal should be stated more clearly and appear earlier. Consider including a hypothesis or a statement of expected outcomes.
- Materials and Methods:
- Emphasize at the very beginning that only one individual kodkod was used for oocyte collection.
- For domestic cat controls, clarify the replication strategy: specify whether "N" refers to the number of oocytes, cumulus-oocyte complexes (COCs), or individual animals used at each step of the experiments. This ambiguity is present throughout the methods.
- Ensure consistency in the level of detail provided for different methods. Some protocols are described excessively, while others are too brief. Maintain a balance appropriate for a scientific journal.
- Revise the descriptive style. The methods currently read more like a series of laboratory recipes. Reframe them to describe the experimental design and the rationale behind each step more clearly.
- Regarding statistical analysis, beyond mentioning "t-test" and "Kruskal-Wallis test," provide a clear justification for why each specific test was chosen for each dataset analyzed.
- Results:
- The results section is currently overloaded with long paragraphs, some of which contain methodological details that should be in the M&M section.
- Explicitly state the small sample size (e.g., number of oocytes from the single individual) at appropriate points in the results to contextualize the findings.
- Question the reporting of p-values for statistical tests given the extremely small sample size, particularly for kodkod oocytes. Discuss the implications of these small numbers rather than implying statistical significance where it might be questionable.
- Avoid reporting numerous "mean ± SD" values directly within the text. Present these data clearly in well-designed tables or graphs to improve readability.
- Clearly and prominently highlight the main discovery: that kodkod oocytes can mature and develop into blastocysts, while also acknowledging the very low efficiency observed.
- Figures:
- Avoid simply restating information that is already evident from the figures in the text. Instead, interpret the results presented in the figures.
- Describe observed trends, correlations, or notable patterns.
- Clearly point out and highlight the unique findings or significant aspects within each figure, rather than letting them be obscured by descriptive text.
- Discussion:
- The discussion largely reiterates the results without sufficient interpretation or synthesis.
- Address the repetitive comparisons to other felids. Instead of just stating similarities/differences, discuss the implications of these comparisons for understanding kodkod reproductive biology and for adapting ART protocols.
- Crucially, include a comprehensive discussion of the limitations of the current research, particularly the single individual and small sample size, and how these limitations might affect the interpretation and generalizability of the findings.
- Articulate clearly what new knowledge these results contribute to the existing understanding of reproductive biology in domestic cats and other felids.
- Expand the discussion on the conservation applications of this research and outline specific directions for future research. These aspects are currently discussed too briefly compared to other topics (e.g., zona pellucida thickness, lipid content, transcriptomic differences).
- Conclusions:
- The conclusion section largely restates information already presented in the results and discussion.
- Reframe this section to clearly identify the knowledge gap that this study addresses.
- Explicitly mention the study's limitations as part of the conclusion.
- Clearly state what makes these results novel and how they advance existing knowledge compared to previous studies (even if none existed for kodkods, frame it against broader felid research).
- Conclude with clear and actionable recommendations for practical next steps in future research to further support kodkod conservation.
- References:
- While acknowledging that the 12 self-citations are relevant, ensure that the text flows naturally and does not appear to be overly reliant on internal group publications where broader literature might also be equally or more pertinent. Review the surrounding text for any opportunities to integrate other relevant references.
Author Response
Comments 1:
- The manuscript presents groundbreaking information on kodkod oocyte biology and development, which is highly valuable for conservation. However, the presentation often obscures the novelty and significance of these findings.
- A critical limitation throughout the study is the use of a single kodkod individual and a small number of oocytes. This aspect needs to be explicitly and consistently highlighted and discussed in all relevant sections (Abstract, M&M, Results, Discussion, Conclusion).
- The overall narrative and logical flow need to be significantly improved across all sections to make the manuscript more coherent and accessible, particularly for an interdisciplinary audience that may include conservationists.
Response 1:
Thank you for the observations. We are fully aware that one of the main limitations of this study is the number of oocytes, which could not be higher since they were obtained from a single individual that unfortunately died at the wildlife rehabilitation center of the university. In this context, it is not feasible to increase the number of oocytes as is commonly done in studies involving domestic species. The kodkod is a highly vulnerable and endangered species, and its handling for this type of research is restricted. Collecting additional oocytes would require laparoscopic or ovariohysterectomy procedures in several individuals, which would clearly contravene bioethical and animal welfare standards, as well as negatively impact the reproductive potential of these animals, an outcome contrary to the very purpose of this research, which aims to enhance kodkod reproduction and conservation.
As you pointed out, in this revised version of the manuscript we have explicitly stated that the scope of our findings is limited by the small sample size, and that further studies including a greater number of individuals are required, provided that bioethical and conservation principles are strictly respected. Taking these aspects into account, this manuscript describes for the first time the protocols that could be used for in vitro maturation and embryo generation in the kodkod, which were successfully demonstrated and can be
Additionally, it is important to consider the ethical and animal welfare implications of sample collection in wild species, particularly in species such as the kodkod, where it would not be ethically appropriate to collect an excessive number of samples merely to increase the sample size. This must be understood in the context of a species with a declining population, where obtaining reproductive samples from a large number of individuals would be counterproductive to the main goal of these studies, which is precisely to support their reproduction and conservation.
Comments 2:
Simple Summary:
-
- The current summary contains too much specialized terminology without adequate explanation for a non-specialist audience.
- It reads more like a condensed abstract rather than a simple, accessible summary.
- Please explicitly highlight the connection between this research and the protection/conservation of the kodkod species.
- Move detailed methodological descriptions to the abstract and/or methods section.
- Emphasize the unique opportunity and importance of working with oocytes from an endangered species.
- Provide brief, clear explanations for terms such as "blastocyst" and "polar bodies."
- The conclusive sentence needs to be rewritten to better convey the significance of this work as a foundational step for future kodkod conservation efforts.
- Response 2: We are thankful for the positive comments of the reviewer. The Simple Summary was completely rewritten, taking into account the comments from Reviewers 1 and 2. It was written in less technical language, emphasizing the importance of conservation and highlighting the relevance of these results for the preservation of the kodkod.
Comments 3:
Abstract:
-
- The abstract currently blends methods, results, and conclusions, leading to a fragmented and difficult-to-follow narrative.
- Revise the logical flow to clearly separate background, objectives, methods overview, key results, and main conclusions.
- Avoid overly long sentences packed with multiple ideas. Strive for conciseness and clarity.
- Remove exact numerical data (mean ± SD) from the abstract; these details belong in the results section. Focus on the main findings and their significance.
- Crucially, explicitly state and emphasize that only one individual kodkod and a very limited number of oocytes (e.g., 13 suitable for maturation) were used. This is a major limitation that must be transparently acknowledged upfront.
- The current conclusion is too general. Clearly articulate the specific contribution of this research and explain why these initial results are important beyond simply being the first report.
- Response 3: Thank you. The abstract was rewritten following the reviewer´s suggestions. Numerical data was removed, and the flow of the text was modified. We have emphasized that only one pair of ovaries and oocytes from one individual were used. Additionally, the relevance of this study along with future perspectives was clearly indicated.
Comments 4:
Introduction:
-
- The initial broad review of wild felids is too extensive and dilutes the focus on the kodkod. Begin the introduction by immediately highlighting the endangered status, population decline, and threats faced by the kodkod.
Response: The text was modified following the reviewer’s comments.
- Address the repetition of "assisted reproductive technologies" (ART). Organize the information more effectively, perhaps introducing ART generally, then focusing on specific techniques relevant to this study.
Response: Thank you. The information was organized, and redundancy of concepts was corrected.
- Some references regarding reproductive technologies in felids appear to reiterate similar ideas. Streamline these discussions to avoid redundancy.
Response: Thank you. The text was rewritten avoiding redundancy.
- The introduction currently reads too much like a general review of methods. Focus on building the specific context for this study.
Response: We agree, and we have rewritten the text following the reviewer´s comments.
- Clearly articulate the existing knowledge gap regarding kodkod reproduction early in the introduction. State explicitly that "little is known on kodkod reproduction.
Response: This information was addressed early in the introduction, where the limited knowledge of kodkod reproduction was explicitly stated.
- Strengthen the link between reproductive technologies and their role in kodkod conservation strategies.
Response: Thank you. This information has been included in the revised version of the manuscript.
- The research goal should be stated more clearly and appear earlier. Consider including a hypothesis or a statement of expected outcomes.
Response: Thank you for this important point to improve our manuscript. The link between ARTs and kodkod conservation was included in the text. A hypothesis was included.
Comments 5:
Materials and Methods:
-
- Emphasize at the very beginning that only one individual kodkod was used for oocyte collection.
- Response: Thank you. We have highlighted that our study was conducted based on the experiments done in one individual. We have emphasized that we have worked with one pair of ovaries from one kodkod individual.
- For domestic cat controls, clarify the replication strategy: specify whether "N" refers to the number of oocytes, cumulus-oocyte complexes (COCs), or individual animals used at each step of the experiments. This ambiguity is present throughout the methods.
- Response: Thank you. “N” was explained in the methods section and in the tables of the results section.
- Ensure consistency in the level of detail provided for different methods. Some protocols are described excessively, while others are too brief. Maintain a balance appropriate for a scientific journal.
- Response: Thank you. We have revised the methodology and established a consistency along the manuscript, maintaining the flow of a scientific article. The more complex procedures, such as IVM and IVF, were described in detail because they have several steps compared for example to the morphological analysis, which mainly involves the evaluation of images. For this reason, some protocols are more extensive while others are more concise.
- Revise the descriptive style. The methods currently read more like a series of laboratory recipes. Reframe them to describe the experimental design and the rationale behind each step more clearly.
- Response: The protocols you are referring to correspond to the preparation of culture media for the different processes involved in in vitro embryo production. This study is focused on in vitro embryo production; therefore, the protocols must be described in detail to ensure replicability. Moreover, it is important to emphasize that these protocols were originally developed for the domestic cat and were applied here to oocytes from an endangered species, for which no previous records exist. The experimental design was described in detail and also presented as an explanatory diagram at the beginning of the Materials and Methods section. This information will not be repeated, as it would be redundant.
- Regarding statistical analysis, beyond mentioning "t-test" and "Kruskal-Wallis test," provide a clear justification for why each specific test was chosen for each dataset analyzed.
- Response: The corrections were made. Justification of each specific test was added to the statistical analysis description.
Comments 6:
- Results:
- The results section is currently overloaded with long paragraphs, some of which contain methodological details that should be in the M&M section.
- Response: Methodological details were moved to M&M section
- Explicitly state the small sample size (e.g., number of oocytes from the single individual) at appropriate points in the results to contextualize the findings.
- Response: This was added in several paragraphs and across different sections throughout the manuscript.
- Question the reporting of p-values for statistical tests given the extremely small sample size, particularly for kodkod oocytes. Discuss the implications of these small numbers rather than implying statistical significance where it might be questionable.
- Response: This is not an extremely small sample size for a morphological analysis of oocytes, if this were considered an extremely small sample size, please indicate what you consider to be an appropriate sample size. For morphological analysis each oocyte is considered as an individual, we analyzed the oocytes and not the subject. The fact that only one kodkod was used in this study is not necessarily related to the p-values or the significant differences found, because the unit of analysis is the oocyte, not the animal from which they were obtained. The p-values were included to demonstrate the validity of this analysis and the differences observed, and they will not be removed from this manuscript.
- Avoid reporting numerous "mean ± SD" values directly within the text. Present this data clearly in well-designed tables or graphs to improve readability.
- Response: This has been corrected. Please indicate specifically where in the text these data are described in detail for a more thorough review, as they were not found to be extensively presented.
- Clearly and prominently highlight the main discovery: that kodkod oocytes can mature and develop into blastocysts, while also acknowledging the very low efficiency observed.
- Response: This observation was added.
Comments 7:
Figures:
- Avoid simply restating information that is already evident from the figures in the text. Instead, interpret the results presented in the figures.
- Describe observed trends, correlations, or notable patterns.
- Clearly point out and highlight the unique findings or significant aspects within each figure, rather than letting them be obscured by descriptive text.
Response: I agree. These comments were taken into consideration, and the modifications in the results section and figures have been made.
Comments 8:
Discussion:
- The discussion largely reiterates the results without sufficient interpretation or synthesis.
- Address the repetitive comparisons to other felids. Instead of just stating similarities/differences, discuss the implications of these comparisons for understanding kodkod reproductive biology and for adapting ART protocols.
Response: This was already done, indicating the importance of these results for the improvement of kodkod embryo culture and embryo production by iSCNT.
- Crucially, include a comprehensive discussion of the limitations of the current research, particularly the single individual and small sample size, and how these limitations might affect the interpretation and generalizability of the findings.
- Articulate clearly what new knowledge these results contribute to the existing understanding of reproductive biology in domestic cats and other felids.
- Expand the discussion on the conservation applications of this research and outline specific directions for future research. These aspects are currently discussed too briefly compared to other topics (e.g., zona pellucida thickness, lipid content, transcriptomic differences).
Response: The discussion it has been rewritten in accordance with these comments.
Comments 9:
Conclusions:
- The conclusion section largely restates information already presented in the results and discussion.
- Reframe this section to clearly identify the knowledge gap that this study addresses.
- Explicitly mention the study's limitations as part of the conclusion.
- Clearly state what makes these results novel and how they advance existing knowledge compared to previous studies (even if none existed for kodkods, frame it against broader felid research).
- Conclude with clear and actionable recommendations for practical next steps in future research to further support kodkod conservation.
Response: Thanks for the observations made. Conclusions have been modified in order to these comments.
References:
-
- While acknowledging that the 12 self-citations are relevant, ensure that the text flows naturally and does not appear to be overly reliant on internal group publications where broader literature might also be equally or more pertinent. Review the surrounding text for any opportunities to integrate other relevant references.
Response: The number of self-citations has been reduced in this version.
Round 2
Reviewer 1 Report
Comments and Suggestions for Authors
I think all my comments have been adequately addressed. The revised manuscript shows improvement. A careful proofreading is required before publication.
Author Response
Thank you very much for your feedback. I appreciate your positive evaluation and I’m glad to know that the revised manuscript shows improvement. We carefully proofread the entire text once more to ensure it is fully ready for publication.
Reviewer 2 Report
Comments and Suggestions for Authors
The revised version of the manuscript shows substantial improvement over version 1. The authors met the majority of reviews comments. The structure and logic have improved. Methods section is consistent, statistical tests were named and justified.The text describes trends and quality. The discussion section was reorganised and now highlights species conservation and outlines future research. This manuscript is the first detailed description of oocyte collection, in vitro maturation, fertilisation and early embryonic development of endangered kodkod species. The authors used protocols adapted for domestic cat assisted reproductive technologies. The results have shown that kodkod might develop early embryos in vitro. These meaningful results represent the first step towards technologies of endangered species conservation.
The revised version is clear and scientifically sound. The results of research provide unique data, filling the knowledge gap in Kodkod early development. The manuscript is almost ready for publication.
Accept in present form.
Author Response
Thank you very much for your thorough and positive evaluation. We truly appreciate your recognition of the improvements made to the manuscript and your encouraging comments regarding its scientific quality and contribution to the conservation of the endangered kodkod. We are grateful that you consider the study clear, consistent, and valuable in filling the existing knowledge gap in early embryonic development of this species. We sincerely thank you for recommending the manuscript for publication and for your constructive feedback throughout the review process.